# Phase I Study of a B Cell-Based and Monocyte-Based Immunotherapeutic Vaccine, BVAC-C in Human Papillomavirus Type 16- or 18-Positive Recurrent Cervical Cancer

**DOI:** 10.3390/jcm9010147

**Published:** 2020-01-05

**Authors:** Chel Hun Choi, Hyun Jin Choi, Jeong-Won Lee, Eun-Suk Kang, Duck Cho, Byung Kwan Park, Yong-Man Kim, Dae-Yeon Kim, Hyungseok Seo, Myunghwan Park, Wuhyun Kim, Ki-Young Choi, Taegwon Oh, Chang-Yuil Kang, Byoung-Gie Kim

**Affiliations:** 1Department of Obstetrics and Gynecology, Samsung Medical Center, Sungkyunkwan University School of Medicine, Seoul 06351, Korea; chelhun.choi@samsung.com (C.H.C.); garden.lee@samsung.com (J.-W.L.); 2Department of Obstetrics and Gynecology, Chung-Ang University hospital, College of medicine, Chung-Ang University, Seoul 06974, Korea; Hj0818.choi@gmail.com; 3Department of Laboratory Medicine and Genetics, Samsung Medical Center, Sungkyunkwan University School of Medicine, Seoul 06351, Korea; esk.kang@samsung.com (E.-S.K.); duck.cho@samsung.com (D.C.); 4Department of Radiology, Samsung Medical Center, Sungkyunkwan University School of Medicine, Seoul 06351, Korea; bk1436.park@samsung.com; 5Department of Obstetrics and Gynecology, University of Ulsan College of Medicine, Asan Medical Center, Seoul 06351, Korea; ymkim@amc.seoul.kr (Y.-M.K.); kdyog@amc.seoul.kr (D.-Y.K.); 6Laboratory of Immunology, Department of Molecular Medicine and Biopharmaceutical Sciences, Graduate School of Convergence Science and Technology, Seoul National University, Seoul 06351, Korea; hseo@lji.org (H.S.); cykang@snu.ac.kr (C.-Y.K.); 7Cellid, Inc., Seoul 06351, Korea; mh.parkk@gmail.com (M.P.); whkim@cellid.co.kr (W.K.); kychoi@cellid.co.kr (K.-Y.C.); tgoh@cellid.co.kr (T.O.)

**Keywords:** B cell, monocyte, therapeutic vaccine, HPV 16, HPV 18, cervical cancer

## Abstract

BVAC-C is a B cell-based and monocyte-based immuno-therapeutic vaccine transfected with a recombinant human papillomavirus (HPV) 16/18 E6/E7 gene and loaded with alpha-galactosyl ceramide, which is a natural killer T cell ligand. This phase I study sought to determine the tolerability and immunogenicity of BVAC-C in platinum-resistant recurrent cervical cancer patients. Patients with HPV 16-positive or 18-positive recurrent or persistent cervical cancer who had received at least one prior platinum-based combination chemotherapy were enrolled. BVAC-C was injected intravenously three times every four weeks, and dose escalation was planned in a three-patient cohort design at doses of 1 × 10^7^, 4 × 10^7^, or 1 × 10^8^ cells/dose. Eleven patients were enrolled, and six (55%) patients had received two or more lines of platinum-based chemotherapy prior to enrollment. Treatment-related adverse events (TRAEs) were observed in 21 cycles. Most TRAEs were mild fever (*n* = 6.55%) or myalgia (*n* = 4.36%). No dose-limiting toxicities occurred. The overall response rate was 11% among nine patients evaluable, and the duration of response was 10 months. Five patients (56%) achieved a stable disease for 4.2–11 months as their best overall response. The median progression-free survival in all patients was 6.8 months (95% CI, 3.2 to infinite months), and the overall survival rate at 6 and 12 months was 89% (95% CI, 71 to 100%) and 65% (95% CI, 39 to 100%), respectively. BVAC-C induced the activation of natural killer T cells, natural killer cells, and HPV 16/18 E6/E7-specific T cells upon vaccination in all patients evaluated. BVAC-C was well tolerated and demonstrated a durable anti-tumor activity with an immune response in HPV 16-positive or 18-positive recurrent cervical carcinoma patients. A Phase 2 efficacy trial is currently underway.

## 1. Introduction

Although screening programs and prophylactic vaccines have dramatically reduced the incidence of cervical cancer, treatment options for patients with metastatic or recurrent disease are very limited [1,2]. Even though adding bevacizumab to first-line combination chemotherapy was shown to provide an overall survival (OS) advantage [3], only limited treatment options exist after progression with this treatment, such as single-agent therapies and palliative care [4]. Therefore, novel therapeutic modalities are needed to improve the prognosis of patients with recurrent cervical cancer.

Cancer immunotherapy has been a recent focus in clinical oncology [5,6]. Recently, pembrolizumab was approved in the US for PD-L1-positive recurrent cervical carcinoma for which first-line platinum chemotherapy failed. However, the response rate was only 15%, even in PD-L1-positive cases, and the median progression-free survival (PFS) of all patients was 2.1 months [7], which indicates that there is opportunity for improvement. Virtually all cervical cancers are causally linked to HPV infections [8], and HPV E6 and E7 oncoproteins have been immunotherapy targets for HPV-induced cancers [9]. Among the immunotherapeutic strategies, a therapeutic vaccine targeting HPV E6/E7 could potentially be used to treat HPV-positive cervical carcinoma [10,11]. To this end, various approaches have been tested, such as chimeric virus-like particles, a dendritic cell vaccine, fusion proteins, peptides, an attenuated virus, and HPV-specific T-cell therapy [9,12,13].

BVAC-C is the first pipeline to utilize CeliVax^TM^, a first-in-class B cell-based and monocyte-based cancer therapeutic vaccine technology that activates both innate and adaptive immune effectors [14,15]. It consists of autologous B cells and monocytes as antigen presenting cells that are transfected with highly antigenic recombinant genes derived from the viral oncogenes E6 and E7 of HPV types 16 and 18 and loaded with α-galactosyl ceramide, which is a natural killer T cell ligand. Furthermore, BVAC-C is manufactured overnight starting with the collection of cells from a patient. Therefore, this vaccine can potentially target any cancer that is HPV type 16-positive or 18-positive, including cervical cancer. Cancer cells resist the attack of cytotoxic T cells by a gradual loss of surface antigens and/or MHC molecules. BVAC-C can eliminate those immune-escaping cancer cells by activating innate effector cells such as natural killer T (NKT) cells and natural killer (NK) cells. In addition, cancer cells resist the action of T cells and natural killer cells by inducing an exhaustion of those effector cells in the tumor microenvironment. This can be counteracted by the BVAC-C vaccination, which activates NKT cells in the tumor microenvironment to release cytokines such as IL-21, IL-2, and IFN-γ.

In this case, we report the results of an open-label, dose-escalation, multiple dosing phase I study to evaluate the safety, tolerability, immune response, and preliminary efficacy of BVAC-C in patients with metastatic or recurrent HPV type 16-positive or 18-positive cervical cancer after the failure of standard platinum-based chemotherapy.

## 2. Methods

### 2.1. Patients and Eligibility Criteria

This study was a multicenter, Phase I, single-arm trial investigating the safety and efficacy of BVAC-C in metastatic or recurrent cervical cancers. The Institutional Ethics Review Board of the Samsung Medical Center (No. 2016-04-112) and Asan Medical Center (No. 2017-0949) reviewed and approved the study protocol, which was registered in the Clinical Trials Registry as NCT02866006. Informed consent was obtained for all patients, and the trial was carried out in accordance with the Helsinki declaration on experimentation on human subjects. The primary end points were safety and tolerability, and the secondary objectives were to evaluate the immunologic response and preliminary efficacy.

Patients were eligible if they had histologically confirmed recurrent HPV 16/18-positive cervical cancer and had progressed after prior standard platinum-based chemotherapy with or without bevacizumab. The HPV type was determined using an archived formalin-fixed paraffin-embedded tumor sample or a newly obtained tumor biopsy sample when HPV typing results of the last five years were not available. Other inclusion criteria were an Eastern Cooperative Oncology Group performance status (ECOG) of 0–2, older than 20 years of age, a life expectancy of more than 3 months, measurable disease per immune-related response criteria (irRC), adequate bone marrow function (absolute neutrophil count ≥ 1500/mm^3^ and platelet count ≥ 100,000/mm^3^), adequate renal function (serum creatinine ≤ 2.5 mg/dL), and adequate liver function (transaminase within 2.5 times the institution’s upper limit of normal). Patients were excluded if they had received chemotherapy, radiotherapy, or immunotherapy within four weeks before treatment initiation, had severe ischemic heart disease, had an active infectious disease requiring systemic therapy, had active central nervous system metastases, or had a steroid-dependent autoimmune disease.

### 2.2. Preparation of BVAC-C

Peripheral blood mononuclear cells (PBMC) were collected by apheresis from each patient at participating hospitals, and the PBMC were transported to GMP facility on the same day. Red blood cells were lysed using hypotonic buffer, and T cells were depleted by magnetic cell separation technology (Miltenyi, Bergisch Gladbach, Germany). The cells remaining were transduced by replication of a defective adenovirus harboring an HPV E6 and E7 recombinant gene, and incubated overnight with α-galactosyl ceramide. The cells were harvested, formulated in freezing media, and then filled in cryogenic vials. The vials of cells were stored in vapor nitrogen until release.

### 2.3. Treatment and Assessments

Patients received BVAC-C as an intravenous injection every four weeks for a total of three doses, as described in the study protocol. Dose escalation was performed in three patient cohorts with doses of 1 × 10^7^, 4 × 10^7^, and 1 × 10^8^ cells/dose. Treatment was discontinued in the case of withdrawal of consent, unacceptable toxicity, or investigator decision. Safety was monitored throughout the study and for 30 days after treatment discontinuation. All adverse events were graded according to the National Cancer Institute’s Common Terminology Criteria for Adverse Events (version 4.0). Dose-limiting toxicity (DLT) was defined as a cytokine release syndrome of grade 3 or higher or any adverse drug reaction of ≥grade 3. The relationship between an adverse event and the investigational agent was determined by the investigator.

The secondary end points were a disease control rate (DCR) and an overall response rate (ORR). Response was assessed by computed tomography or magnetic resonance imaging every 12 weeks, and each measurable lesion was evaluated according to irRC by a diagnostic radiologist (BKP). If radiologic imaging indicated progressive disease, a repeat assessment was required four weeks later to confirm progression. Post-study follow-up data on survival were gathered by the study investigators.

### 2.4. Immunological Response

#### 2.4.1. Ex Vivo IFN-γ ELISPOT

Peripheral blood mononuclear cells (PBMCs) were collected and cryopreserved from each participant at the screening visit, visit 2 to final visit (Appendix A). Thawed PBMCs were incubated with X-VIVO (Lonza, Köln, Germany) for more than 16 h at 37 °C, 5% CO_2_. PBMCs (2 × 10^5^ cells per well) were subsequently stimulated with 2 mg/mL of two different pools of HPV16 E6-derived and E7-derived peptides (15-mer with eight overlapping amino acids) and HPV18 E6-derived and E7-derived peptides (15-mer with 8 overlapping amino acids) for 48 h. After stimulation, spots indicating IFN-γ-secreting cells were developed according to the manufacturer’s instructions (R&D systems, Minneapolis, MN, USA). HPV-specific T-cell responses were calculated by subtracting the spot numbers of peptides unstimulated T cells from those of peptide stimulated T cells.

#### 2.4.2. Serum-Cytokine Analysis

Activated NK and NKT cells secrete IFN-γ and TNF-α. Hence, we used the levels of these cytokines in the patient blood samples as activation markers of NK and NKT cells. IFN-γ and TNF-α were measured in human plasma (Visit 2, Visit 2 + 1 day, Visit 4 + 1 day, Visit 6 + 1 day) using Milliplex technology (Millipore, Burlington, MA, USA). Assays were performed using a Luminex 200 microbead analyzer (Millipore, Burlington, MA, USA), according to the manufacturer’s instructions.

#### 2.4.3. Intracellular Staining

Cryopreserved and thawed PBMCs were resuspended in X-VIVO (Lonza, Köln, Germany), and rested for 21 h at 37 °C, 5% CO_2_. PBMCs were plated in duplicate and added with secretion inhibitors (Golgi-plug/Golgi-stop, BD Bioscience, Franklin Lakes, NJ, USA) 1 h after plating. After resting, cells were washed with PBS for subsequent immunostaining and polychromatic flow cytometric analysis. Intracellular staining for cytokines was performed after surface staining with the Cytofix/Cytoperm kit (BD Bioscience, Franklin Lakes, NJ, USA). Antibodies for surface staining were human CD3 (BV786, BD Bioscience, Franklin Lakes, NJ, USA), human CD16 (APC, Biolegend, San Diego, CA, USA). Antibodies for intracellular staining were human IFN-γ (PE/Cy7, Biolegend, San Diego, CA, USA). Fluorescence-activated cell sorting analysis was accomplished using an LSR Fortessa flow cytometer (BD Bioscience, Franklin Lakes, NJ, USA), and data were analyzed using FlowJo software (FlowJo, LLC, Ashland, OR, USA).

### 2.5. Statistical Analysis

Descriptive statistics were used to assess the safety and tolerability endpoints in all patients who received at least one dose of BVAC-C. Immunogenicity endpoints were assessed in the intent-to-treat population, defined as all subjects who received any amount of study treatment and who had at least one post baseline immunological evaluation. Efficacy was assessed in all patients who had measurable disease at baseline per RECIST v1.1 and who received all three doses of BVAC-C. The last follow-up for this analysis was 1 January 2019.

## 3. Results

### 3.1. Patients

Of the 18 patients with recurrent or persistent cervical cancer that we screened, 11 patients were enrolled and received vaccine treatment between September 2016 and January 2018. Seven patients were ineligible because of a non-HPV 16 or 18 type (*n* = 4), biochemical abnormalities (*n* = 2), or an active hepatitis B infection (*n* = 1). The characteristics of these patients are shown in Table 1. All but one patient had an ECOG performance status of 1. All patients presented with metastatic disease, which was most frequently located in the lung (*n* = 6.55%), lymph nodes (*n* = 5.45%), pelvis (*n* = 4.36%), and/or liver (*n* = 2.18%), and six patients (55%) had metastatic disease at multiple sites. Six (55%) patients had received two or more lines of platinum-based chemotherapy for advanced disease prior to the study. Among the patients enrolled in this study, nine (82%) were HPV 16-positive and two (18%) were HPV 18-positive.

### 3.2. Adverse Events

All adverse events (AEs) of any grade observed in the study included ninety events (Appendix A). The most frequent AEs observed during this treatment were pyrexia (*n* = 7; 63%), anemia (*n* = 7; 63%), and myalgia (*n* = 6; 54%). These AEs were all manageable. Treatment-related adverse events (TRAEs) are summarized in Table 2. TRAEs were observed in 21 cycles, and they were a mild fever (*n* = 6.55%), myalgia (*n* = 4.36%), vomiting (*n* = 1.9%), headache (*n* = 1.9%), chills (*n* = 1.9%), diarrhea (*n* = 1.9%), cytokine release syndrome (*n* = 1.9%), and fatigue (*n* = 1.9%). No grade 3 or 4 TRAEs were observed. No patient discontinued trial participation due to unacceptable toxicities, and no dose-limiting toxicities occurred. No deaths with a possible relation to the study therapy were noted. The deaths reported were related to the progression of the underlying tumor.

### 3.3. Tumor Response

Nine patients were evaluable for a clinical response (Table 3). The ORR was 11% (95% CI, 0% to 32%), and five patients achieved a stable disease (56%, 95% CI, 23% to 88%, SD). The DCR was 67% (6/9), and the duration of disease control was 4.2–11 months. Representative cystic (necrotic) changes in computed tomography (CT) images without a change in tumor size was shown in Figure 1A. Three patients developed progressive disease (33%, 95% CI, 3% to 64%, PD) at the time of the first evaluation. Two patients out of three PD cases were found to have cervical adenocarcinoma with peritoneal carcinomatosis. A decrease in the diameter of target lesions from baseline was observed in three (33%) of nine evaluable patients by a CT scan (Figure 1B,C). One patient with partial response (PR) was maintained over time. Swimmer plots provided the potential persistence of disease control even without ongoing treatment. The median PFS in all patients (except Patient S31) was 6.8 months (95% CI, 3.2 to infinite months), and the PFS rate at 6 months was 56% (95% CI, 31 to 100%) (Figure 1D). The median OS was 12 months (95% CI, 12 to infinite months), and the OS rate at 6 and 12 months was 89% (95% CI, 71 to 100%) and 65% (95% CI, 39 to 100%), respectively.

### 3.4. Immunogenicity

#### 3.4.1. Serum-Cytokine Analysis and Intracellular Staining

PBMCs and plasma were collected from each patient to evaluate both innate and adaptive immune responses (Figure 2A). BVAC-C induces the activation of NKT cells via α-galactosyl ceramide, which is loaded on CD1d molecules of B cells and monocytes in the drug product. The activation of NKT cells leads to IFN-γ secretion and stimulates NK cells. Assuming that NK and NKT cells were activated within 24 h after BVAC-C treatment based on non-clinical studies, we assessed the blood concentrations of several cytokines one day after BVAC-C treatment to evaluate NK and NKT cell activity (Figure 2B, Table 3). One day after BVAC-C treatment, all patients had elevated levels of IFN-γ (29.7 ng/L to 441.9 ng/L) and TNF-α (18.4 ng/L to 281.7 ng/L) compared to the levels at baseline (Table 3). The fold increases of the IFN-γ level varied within the range of 1.7-fold (S02) to 181.8-fold (S31) after BVAC-C administration while those of TNF-α levels varied from 2.2-fold (S02) to 10.0-fold (S31) above baseline. In addition, NK cells in the peripheral blood from Patient S02 were activated 7.1-fold one day after the second dose of BVAC-C (Figure 3B).

#### 3.4.2. Ex Vivo IFN-γ ELISPOT

The number of HPV-specific T cells in patients’ peripheral blood was monitored every two weeks during the study. Of the eight patients evaluated, the number of HPV-specific T cell spots in two patients, S02 and S08, exceeded 1000 spots, which is 0.1% of T cells in PBMC after BVAC-C administration (Figure 2B, Table 3). More than 200 HPV-specific T cell spots were observed in four patients including S12, S13, S14, and S15. The number of spots was less than 200 in patient S09 and S17. We detected fold increases in the spot number after BVAC-C treatment compared to the baseline control. The highest fold increase in the HPV-specific T cell response after three injections of BVAC-C in each patient was from 1.0-fold (S09) to 36.5-fold (S02) (Table 3). While we observed both HPV type 16 and 18-specific T cell spots in most patients, HPV type 16-specific T cell response dominated in S02 (Figure 3C). HPV type 18-specific T cell responses were higher in S08 and S13 patients whose tumors were HPV type 16 positive.

## 4. Discussion

The results of the current study and the first human study of a B cell-based and monocyte-based vaccine in recurrent cervical cancer can be considered favorable. BVAC-C proved to be safe and tolerable. Mild fever and myalgia were the two most frequent AEs, and no DLT was observed. This vaccine also had a promising antitumor activity with notable immunologic responses in patients for whom appropriate treatment options were not available. One (11%) PR and five (56%) SD were observed among the nine patients evaluated. Central necrosis without a change of tumor size (S14) can imply a potential therapeutic benefit of immunotherapy without an objective response by irRC. Overall survival was extended by more than 20 months in three patients, and these patients are still alive at the time of analysis (January 2019). A median OS of 12 months coupled with a 6-month OS rate of 89% further indicate the potential of BVAC-C for treating metastatic cervical cancer. Swimmer plots provided the potential persistence of disease control even without ongoing treatment, which is an important efficacy metric. A phase IIa study evaluating the efficacy of BVAC-C is currently underway.

Various therapeutic vaccine candidates have been tested, including chimeric virus-like particles, a dendritic cell (DC) vaccine, fusion proteins, peptides, an attenuated virus, and recombinant viruses [9,12,13]. BVAC-C, a B cell-based and monocyte-based vaccine, has practical advantages over DC vaccines. Since DCs are very rare in peripheral blood, it takes time and a labor-intensive expansion process to manufacture these drug products. In contrast, B cells and monocytes are abundant in peripheral blood, and these cells can be collected easily and efficiently by apheresis. The manufacturing process for the BVAC-C drug product takes only one day in a GMP facility, so treatment can begin three days after apheresis. Furthermore, the BVAC-C drug product can be stored frozen and administered to a patient more than 10 times. The immunogenicity and antitumor efficacy of BVAC-C observed in this trial strongly support the modes of action of the drug product that were suggested in animal studies. We previously reported that an invariant NKT cell ligand, α-galactosyl ceramide, loaded in a tumor antigen-expressing B cell-based and monocyte-based vaccine (BVAC) elicited diverse antitumor immune responses [14,15,16]. Furthermore, BVAC can overcome the immune-escaping mechanisms of cancer cells, such as MHC-I molecule loss and an exhaustion of immune effector cells in the tumor microenvironment [17,18].

In cervical cancer, new agents with decreased toxicity are needed. A recent advance in the treatment of cervical cancer was the introduction of bevacizumab, which improved the median OS by four months [3]. However, toxicities such as thromboembolism and gastrointestinal and genitourinary fistula increased [3]. The major adverse events observed during this treatment were anemia and fever, which can both be easily managed. The safety profile of BVAC-C from this study supports its use in heavily treated cancer patients.

All patients in the present study had recurrent disease and were resistant to platinum-based chemotherapy [19]. Although there was only one (11%) clinical response in the present study, five patients achieved SD (56%, 95% CI, 23% to 88%) for 4.2–11 months, which indicates durable effects by the vaccine. Three of six patients who experienced clinical benefits had a longer progression-free interval than the interval prior to BVAC-C, which was one to three months. Furthermore, two of the six patients had a progression-free interval of six months after subsequent treatment, which indicates a synergistic effect with the immune memory of BVAC-C.

Immune checkpoint inhibitors can improve the antitumor activity of T cells by blocking inhibitory signals that hamper the cytotoxic functions of tumor-infiltrating T cells. Pembrolizumab was approved in 2018 as a second-line treatment in US. The ORR was 13%, the DCR was 30%, median PFS was 2.1 months, and median overall survival was 9.4 months in a Phase 2 trial [7]. The DCR of 67% and median OS of 12 months obtained here using BVAC-C are very promising. However, a direct comparison between the two trials is difficult. It is widely accepted that combining immunotherapies with distinct modes of action would result in a more robust antitumor efficacy. BVAC-C induced the activation and expansion of tumor-specific T cells in vivo and activated innate immune effectors, such as NKT cells and NK cells. For example, NK cells in peripheral blood from Patient S02 were activated 7.1-fold one day after the second dose of BVAC-C (Figure 3B). This is a distinct mode of action compared with immune checkpoint inhibitors. Therefore, the combination of BVAC-C and a checkpoint inhibitor could have synergistic effects. We have demonstrated the dramatic synergistic efficacy of α-galactosyl ceramide-loaded antigen-presenting cells and PD-1 blockade in a therapeutic murine tumor model [17]. Therefore, it will be of great interest to test BVAC-C and checkpoint inhibitor combinations in patients with cervical cancer.

The trial endpoint of immunogenicity was met in the study. An HPV antigen-specific T cell response was detected in all patients, even though the magnitude of the response was variable (Figure 2B). The levels of IFN-γ and TNF-α in peripheral blood were increased in all patients, which indicated NKT cell and NK cell activation by BVAC-C administration (Figure 2B). Although no distinct immune profile has been consistently linked to the clinical benefit in this cancer vaccine trial, we observed a possible link between the response, irPR and irSD, and the level of IFN-γ in the peripheral blood one day after each BVAC-C administration (Figure 3C). In addition, the IFN-γ levels were higher in patients who survived longer than 12 months. These indicate that the activation of innate immune effectors, NK cells and NKT cells, can play important roles in controlling tumor growth and in prolonging overall survival. Furthermore, patients who survived longer than 12 months showed a much stronger HPV-specific T cell response compared to other patients (Figure 3C). However, the level of the T cell response was no different between PR-SD and PD groups. Taken together, NK cells and NKT cells may have contributed to tumor growth control and survival gain while T cells exerted a long term effect on elongating overall survival. For example, a high level of IFN-γ response and HPV-specific T cell response were observed in S02 and S08 patients who survived longer than 18 months. Since S02 patient showed a partial response continuing survival, we gathered all the results to find possible clues for BVAC-C efficacy (Figure 3). Upon BVAC-C administration, the IFN-γ level increased, and HPV-specific T cell spots boosted multiple injections (Figure 3C). CD3 positive T cell population in lymphocytes was only 0.1% at baseline, and then increased to 66.3% in a month. It is possible that the T cell deficiency benefited S02 to achieve a partial response, since lymphodepletion enhances the efficacy of chimeric antigen T cell therapy dramatically.

## 5. Conclusions

In conclusion, this study suggests that BVAC-C is well tolerated and has a durable antitumor activity in patients with HPV 16/18-positive recurrent cervical cancer. Based on these promising results, we are now conducting a Phase II multicenter clinical trial comparing BVAC-C alone and BVAC-C with a second-line chemotherapy in women with recurrent cervical cancer. Additionally, a study of BVAC-C in combination with PD-1 or PD-L1 inhibitors is currently being developed.

## Figures and Tables

**Figure 1 jcm-09-00147-f001:**
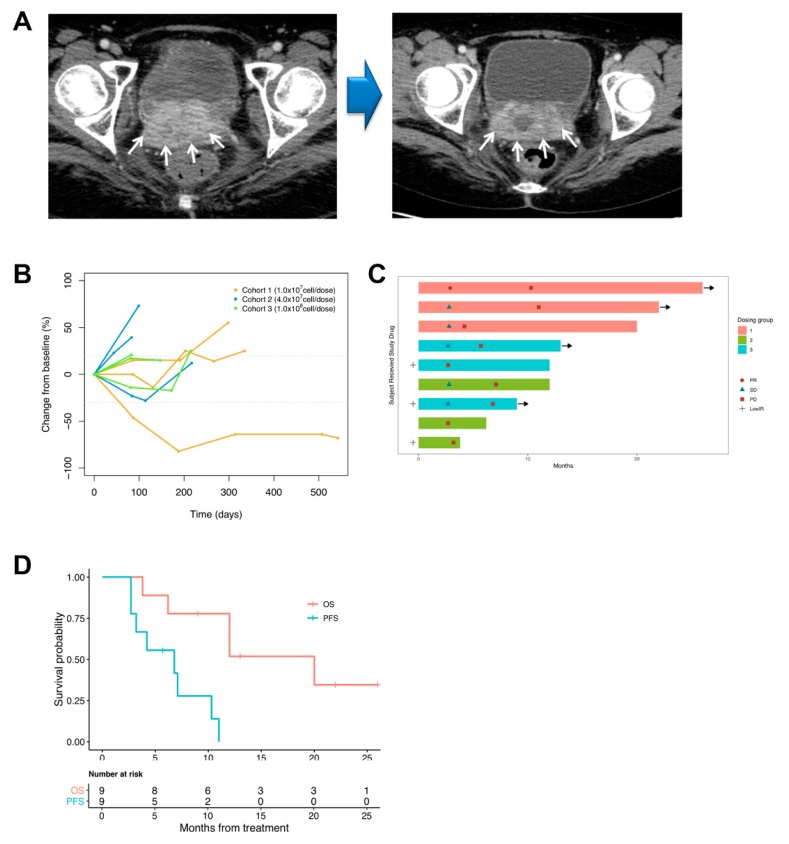
(**A**) Representative CT images during the vaccination protocol. The maximum diameter of the mass was almost stable at follow-up imaging, but it showed central necrosis. White arrows indicate a recurrent tumor. (**B**) Longitudinal change from baseline in tumor size (*n* = 9). Dotted lines at 20% and −30% indicate the percentage change from baseline and represent progressive disease and partial response, respectively, per RECIST v1.1. (**C**) Swimmer plots provide useful information about responses and the potential persistence of these responses even without ongoing treatment. Continuation of response despite immunotherapy discontinuation is an important efficacy metric. Symbols along each bar could be used to represent various relevant clinical events, such as disease progression (PD), stable disease (SD), partial response (PR), or low immune response (LowIR). (**D**) Kaplan-Meier estimates.

**Figure 2 jcm-09-00147-f002:**
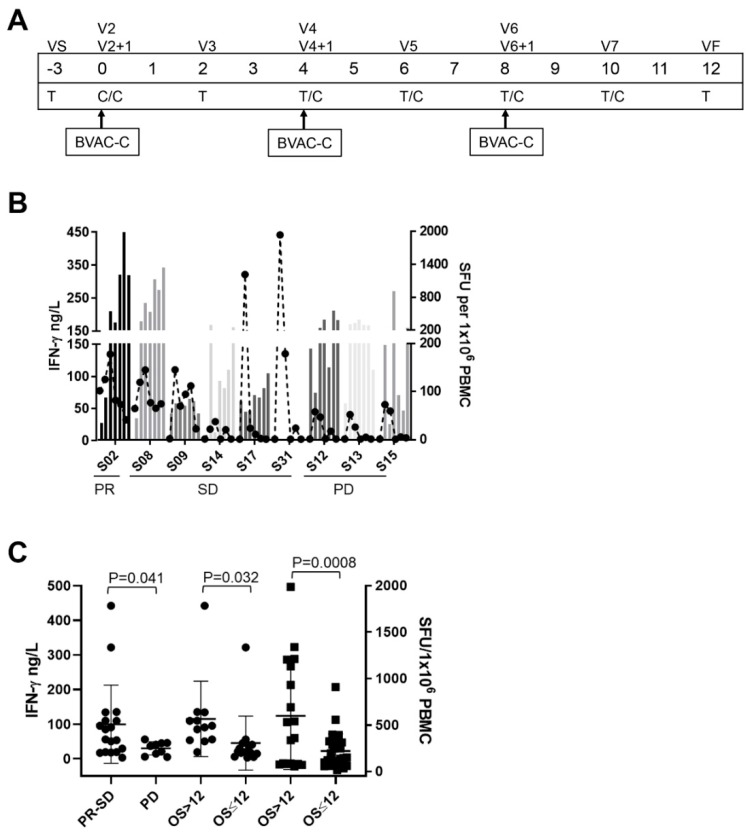
Study schedule and immune response assessment result. (**A**) Study timeline indicating BVAC-C administration and an immune response assessment schedule. T is for T cell Elispot assay, and C is for cytokine measurement. (**B**) Serum IFN-γ levels at each time-points, black circle with a dashed line, and HPV-specific T cell spots at each time-points, grey bars. (**C**) Serum IFN-γ concentrations one day after each BVAC-C administrations are plotted in four groups (circles), and HPV-specific T cell spots of all time-points are plotted in two groups (squares).

**Figure 3 jcm-09-00147-f003:**
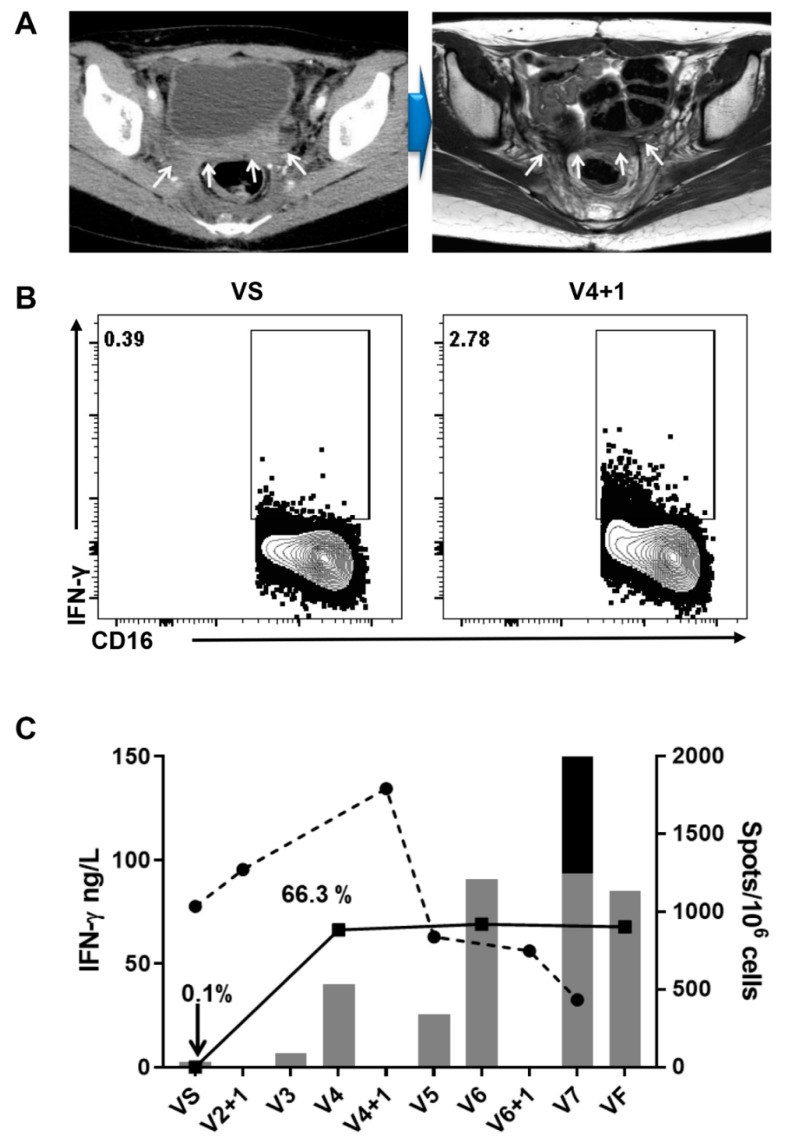
Radiological assessment and immune response analysis of the S02 patient. (**A**) CT scan of the target lesion before BVAC-C administration and MRI scan of the same lesion at VF. (**B**) The activation of natural killer cells in peripheral blood one day after the second administration of BVAC-C. (**C**) Immune response induced by BVAC-C administrations. Serum IFN-γ concentration, black circle with a dashed line, CD3 positive lymphocytes percent, black square with a line, and HPV-specific T cell spots, grey bars for HPV type 16, and black bars for HPV type 18.

**Table 1 jcm-09-00147-t001:** Patient characteristics.

Patient ID	Dose (Cells)	HPV Type	Age	Histology	ECOG PS	* Prior Therapy	Metastases Location	Target Lesion Size (cm)
S02	1 × 10^7^	16	34	SCC	1	1	Pelvis	2.8
S04	1 × 10^7^	18	62	AD	1	2	Lung, Bone	5.1
S08	1 × 10^7^	16	45	SCC	1	1	LNs	1.8
S09	1 × 10^7^	16	50	SCC	1	1	Lung	2.9
S12	4 × 10^7^	16	43	AD	1	3	Lung, Peritoneal	5.6
S13	4 × 10^7^	16	54	SCC	1	1	LNs, Peritoneal	9.2
S14	4 × 10^7^	16	43	SCC	1	2	LNs	7.9
S15	1 × 10^8^	16	35	SCC	1	1	LNs, Liver, Pelvis	9.6
S16	1 × 10^8^	16	35	SCC	1	2	Pelvis, LNs, Lung	7.1
S31	1 × 10^8^	16	62	AD	0	8	Liver, Pelvis, Lung	6.0
S17	1 × 10^8^	18	60	AD	1	5	Lung	1.1

* Prior lines of therapy for advanced disease. Abbreviations: ECOG PS = Eastern Cooperative Oncology Group performance status, SCC = squamous cell, AD = adenocarcinoma, LNs = lymph nodes.

**Table 2 jcm-09-00147-t002:** Treatment-related adverse events of any grade observed in the study (*n* = 11).

Adverse Events	1 × 10^7^ (*n* = 4)			4 × 10^7^ (*n* = 3)			1 × 10^8^ (*n* = 4)			Total, No. (*N* = 11, %)		
	G1	G2	Total	G1	G2	Total	G1	G2	Total	G1	G2	Total
Pyrexia	1	0	1	2	0	2	3	0	3	6(55)	0	6(55)
Chills	0	0	0	0	0	0	1	0	1	1(9)	0	1(9)
Fatigue	0	0	0	0	0	0	1	0	0	1(9)	0	1(9)
Myalgia	0	1	1	1	0	1	2	0	2	3(27)	1(9)	4(36)
Diarrhea	0	0	0	0	0	0	0	1	1	0	1(9)	1(9)
Vomiting	0	0	0	0	0	0	1	0	1	1(9)	0	1(9)
Cytokine release syndrome	0	0	0	0	0	0	0	1	1	0	1(9)	1(9)
Headache	0	0	0	0	0	0	1	0	1	1(9)	0	1(9)

Abbreviations: G1 = Grade 1, G2 = Grade 2, G3 = Grade 3.

**Table 3 jcm-09-00147-t003:** Best overall response as assessed by the investigator review according to irRC (*n* = 9) and immune response induced by BVAC-C administration.

Patient ID	Best Overall Response	PFS	OS	HPV-Specific T Cell ^b^	IFN-γ ng/L ^c^	TNF-α ng/L ^c^
S02	Partial response	10.3	26 ^a^	1202	134.5	18.4
S08	Stable disease	4.2	20	1337	110	62.2
S09	Stable disease	11.0	22 ^a^	55	110.2	48.2
S12	Progressive disease	3.2	3.8	382	45.1	49.1
S13	Progressive disease	2.7	6.2	145	40.8	61.4
S14	Stable disease	7.1	12	255	29.7	53.1
S15	Progressive disease	2.7	12	247	56.2	89.7
S31	Stable disease	5.7	13 ^a^	NA	441.9	281.7
S17	Stable disease	6.8	9 ^a^	137	321.3	56.9

^a^ Denotes censored observations. Survival is ongoing. ^b^ HPV-specific T cell spots/10^6^ PBMC, which was collected at the VF time-point. ^c^ The highest serum IFN-γ or TNF-α concentration measured one day after each BVAC-C administration. Abbreviations: irRC = immune-related Response Criteria. NA = No Assessment.

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
