# Peer review of "Phase I Study of a B Cell-Based and Monocyte-Based Immunotherapeutic Vaccine, BVAC-C in Human Papillomavirus Type 16- or 18-Positive Recurrent Cervical Cancer"

_jcm, 2020, doi:10.3390/jcm9010147_

Round 1

Reviewer 1 Report

Generally, the manuscript is well written, describing the overall aspects of a phase I clinical trial against recurrent cervical cancer.

The method for preparation of this cell-based vaccine is not included in the text and hard to track. The references 14, 15 do not explain these aspects. It is better to describe this point briefly in the materials and methods section with appropriate citation.

The authors tested 3 different doses, without showing the rationale for selecting these doses. The fact that the toxicity was not apparent at the highest dose and the efficacy was still insufficient generates the question what happens at higher doses. The authors should discuss about the doses in the future clinical trials.

Author Response

Reviewer 1

Generally, the manuscript is well written, describing the overall aspects of a phase I clinical trial against recurrent cervical cancer.

The method for preparation of this cell-based vaccine is not included in the text and hard to track. The references 14, 15 do not explain these aspects. It is better to describe this point briefly in the materials and methods section with appropriate citation.

We described the process of manufacturing of BVAC-C in the Methods section

2.2. Preparation of BVAC-C

Peripheral blood mononuclear cells (PBMC) were collected by apheresis from each patient at participating hospitals, and the PBMC were transported to GMP facility on the same day. Red blood cells were lysed using hypotonic buffer, and T cells were depleted by magnetic cell separation technology (Miltenyi, Germany). The cell remained were transduced by replication defective adenovirus harboring HPV E6 and E7 recombinant gene, and incubated overnight with α-galactosyl ceramide. The cells were harvested, formulated in freezing media, and then filled in cryogenic vials. The vials of cells were stored in vapor nitrogen until release.

The authors tested 3 different doses, without showing the rationale for selecting these doses. The fact that the toxicity was not apparent at the highest dose and the efficacy was still insufficient generates the question what happens at higher doses. The authors should discuss about the doses in the future clinical trials.

The maximum dose in the clinical trials (108 cells/dose) was selected based on the result of mouse repeated-dose toxicity testing. The dose we tested in mice was 5x106 cells/head, 4 times injections. We could test up to 109 cells/dose in human by body surface area conversion or up to 5x109 cells/dose by body weight conversion. However, we decided 108 cells/dose as the highest dose because of two reasons. Firstly, it was technically difficult to manufacture higher doses over 108 cells/dose routinely. Secondly, we speculated that higher doses may result in immunological anergy of NKT cells which has been described in many articles (J Clin Invest. 2005;115(9):2572-2583, Nat Immunol. 2002 Sep;3(9):867-74). We believe we have observed tumor-specific immune reaction even with the lowest dose tested, and observed immune anergy to some extent at the highest dose. Therefore we might not test 108 cells/dose in the future clinical trials.

Reviewer 2 Report

This manuscript described the phase 1 study of BVAC-C for HPV 16 or 18 positive recurrent cervical cancer. This therapy is promising immunotherapy for its tolerability and possible antitumor activity. This manuscript is lacking some points as shown below.

Line 31 and 47: HPV type is lacking.

Line 49: It is described ninety cases of AEs. Is it ninety events because this study included 11 cases?

In Methods

The precise manufacturing process for the BVAC-C from apheresis to intravenous administration should be described.

In Figure 3B, PBMCs were stained with anti-CD16 Ab. It should be described in Methods.

Line 323-325: T cells can secret IFN-g, therefore it is difficult to insist that the activation of innate immune effectors can play important roles in controlling tumor growth at this point.

Author Response

Reviewer 2

This manuscript described the phase 1 study of BVAC-C for HPV 16 or 18 positive recurrent cervical cancer. This therapy is promising immunotherapy for its tolerability and possible antitumor activity. This manuscript is lacking some points as shown below.

Line 31 and 47: HPV type is lacking.

Thank you for the comment and I added the type in the text

Line 49: It is described ninety cases of AEs. Is it ninety events because this study included 11 cases?

I have changed to “events”

In Methods

The precise manufacturing process for the BVAC-C from apheresis to intravenous administration should be described.

We described the process of manufacturing of BVAC-C in the Methods section

2.2. Preparation of BVAC-C

Peripheral blood mononuclear cells (PBMC) were collected by apheresis from each patient at participating hospitals, and the PBMC were transported to GMP facility on the same day. Red blood cells were lysed using hypotonic buffer, and T cells were depleted by magnetic cell separation technology (Miltenyi, Germany). The cell remained were transduced by replication defective adenovirus harboring HPV E6 and E7 recombinant gene, and incubated overnight with α-galactosyl ceramide. The cells were harvested, formulated in freezing media, and then filled in cryogenic vials. The vials of cells were stored in vapor nitrogen until release.

In Figure 3B, PBMCs were stained with anti-CD16 Ab. It should be described in Methods.

We added the information of antibodies used in the method section 2.3.3 Intracellular staining

Antibodies for staining surface staining were human CD3 (BV786, BD Bioscience, Franklin Lakes, NJ), human CD16 (APC, Biolegend, San Diego, CA). Antibodies for staining intracellular staining were human IFN-γ (PE/Cy7, Biolegend, San Diego, CA).

Line 323-325: T cells can secret IFN-g, therefore it is difficult to insist that the activation of innate immune effectors can play important roles in controlling tumor growth at this point.

We appreciate this comment. As the reviewer mentioned, T cells as well as innate immune cells (NK and NKT cells) can secrete IFN-g and play crucial roles in controlling tumor growth. NK cell activation status is a useful parameter for testing the efficacy of BVAC-C, which could be examined a day after the administration of BVAC-C. In contrast, T cells need times (a few days) for their activation and IFN-g secretion. We showed not only the activation of NK cells a day after the administration of BVAC-C (Figures 2B and 3B) but also the expansion and activation of T cells following the administration period as shown by cognate peptide restimulation (Figures 2B and 3C). Thus, anti-tumor responses following the administration of BVAC-C might be mediated by the activation of both innate immune cells and T cells.